# HMG20A Inhibit Adipogenesis by Transcriptional and Epigenetic Regulation of MEF2C Expression

**DOI:** 10.3390/ijms231810559

**Published:** 2022-09-12

**Authors:** Ruixiao Li, Shan Meng, Mengting Ji, Xiaoyin Rong, Ziwei You, Chunbo Cai, Xiaohong Guo, Chang Lu, Guoming Liang, Guoqing Cao, Bugao Li, Yang Yang

**Affiliations:** College of Animal Science, Shanxi Agricultural University, Jinzhong 030801, China

**Keywords:** HMG20A, intramuscular fat, C3H10T1/2 cells, adipogenesis, LSD1, MEF2C

## Abstract

Obesity and its associated metabolic disease do serious harm to human health. The transcriptional cascade network with transcription factors as the core is the focus of current research on adipogenesis and its mechanism. Previous studies have found that HMG domain protein 20A (HMG20A) is highly expressed in the early stage of adipogenic differentiation of porcine intramuscular fat (IMF), which may be involved in regulating adipogenesis. In this study, HMG20A was found to play a key negative regulatory role in adipogenesis. Gain- and loss-of-function studies revealed that HMG20A inhibited the differentiation of SVF cells and C3H10T1/2 cells into mature adipocytes. RNA-seq was used to screen differentially expressed genes after HMG20A knockdown. qRT-PCR and ChIP-PCR confirmed that MEF2C was the real target of HMG20A, and HMG20A played a negative regulatory role through MEF2C. HMG20A binding protein LSD1 was found to alleviate the inhibitory effect of HMG20A on adipogenesis. Further studies showed that HMG20A could cooperate with LSD1 to increase the H3K4me2 of the MEF2C promoter and then increase the expression of MEF2C. Collectively, these findings highlight a role for HMG20A-dependent transcriptional and epigenetic regulation in adipogenesis.

## 1. Introduction

Adipose tissue is the main energy storage organ and endocrine organ in animal bodies, and its growth and development seriously affect human health and the economic benefit of livestock and poultry. Studies based on epidemiological studies have shown that obesity and its related metabolic diseases are increasing worldwide, causing 4 million deaths and 120 million disabilities each year [1]. Obesity is caused by excessive energy resulting in an increase in the number and/or size of adipocytes, while abnormal adipogenesis affects the progression of obesity [2,3,4]. Adipogenesis is controlled by a complex regulatory network, including epigenetic modification and multiple transcription factors [5]. Many transcription factors are known to play a role in adipogenesis [6,7,8]. Epigenetics is based on changes in gene expression levels caused by changes in non-gene sequences, including DNA methylation, histone modification, and the regulation of non-coding RNA [9], among which histone modification is one of the most studied epigenetic modification methods [10]. Lysine methylation on the histone H3 protein subunit (H3Kme) is a stable marker for gene expression regulation, and different methylation states confer functional diversity according to their location [11,12].

Intramuscular fat (IMF) is the fat found in the adventitia, fasciculata, and intima of skeletal muscle. It originates from mesenchymal stem cells in the mesoderm. Moderate IMF content can increase sensory characteristics (juiciness, tenderness, and flavor) and the nutritional value of pork [13,14,15]. The screening and identification of various factors affecting IMF content can be helpful for the breeding of commercial pigs. Jiang et al. [16] conducted transcriptome sequencing on the subcutaneous and intramuscular preadipocytes of large white pigs and found that the expression level of HMG20A was significantly increased in the early stage of pig intramuscular preadipocytes differentiation, but whether it was involved in regulating the generation of intramuscular adipocytes remains unknown.

HMG domain protein 20A (HMG20A, also known as iBRAF) is one member of the high mobility group proteins (HMGs) [17], which contains a homologous box (HMG-box) domain located at the amino terminal of the protein and a coiled coil domain located at the carboxyl terminal of the protein. The HMG-box domain can be non-specific binding to DNA, which is of great significance for the realization of protein function [18]. The coiled coil domain at the carboxyl end of proteins is involved in the formation of protein dimers and the composition of complexes [18,19,20]. Gene association analysis showed that an SNPs variation of HMG20A was closely associated with obesity and type 2 diabetes mellitus [21]. Other studies have shown that HMG20A regulates insulin secretion and beta cell mature function in islets [22]. HMG20A binds to the Lys-specific demethylase 1/REST co-repressor 1 (LSD1-CoREST) complex and performs a number of biological functions [23,24], including chromatin modification. During neural differentiation, HMG20A can form a heterodimer with HMG20B, inhibiting the binding of the LSD1–Corest protein complex and increase the trimethylation of lysine 4 on histone H3 protein subunit (H3K4me3) by recruiting methyltransferase MLL, thereby activating the neuronal differentiation program [19,25]. Lysine-specific histone demethylase 1 (LSD1) is the first protein discovered to remove methyl groups from H3K4me1/2 and can act on H3K4 and H3K9 sites [26], and LSD1 has been reported to promote adipogenic differentiation of 3T3-L1 preadipocytes [27].

In this study, we elucidate the inhibitory effect of HMG20A on adipogenesis in SVF cells and C3H10T1/2 cells. Specifically, we showed that HMG20A increases MEF2C expression through transcription, and HMG20A interacts with LSD1 to regulate H3K4me2 levels on the MEF2C gene promoter, affecting MEF2C expression through epigenetic regulation. Our data establish that HMG20A is a multi-faceted regulator that plays a role in inhibiting adipogenesis.

## 2. Results

### 2.1. HMG20A Silencing Promotes Adipogenesis of SVF Cells

To determine the expression characteristics of HMG20A in different tissues and cells, ten tissues including heart, liver, spleen, lung, kidney, subcutaneous fat of back, abdominal subcutaneous fat, longissimus dorsi, psoas, and biceps femoris of DLY pigs were collected. HMG20A is expressed in all 10 tissues (Figure 1A), with the highest expression level in liver followed by muscle. SVF cells were collected at the indicated time points (0, 12 h, 1d, 2d, 3d, 5d, and 7d) during adipogenesis induction in vitro. The key adipogenic gene PPARγ is mainly expressed in the late stage of adipogenic differentiation, with the highest expression at 5d (Figure 1B). However, the expression of HMG20A reached the highest level in 2d (Figure 1C), suggesting that HMG20A may play a role in inhibiting adipogenesis in the early stage of adipogenic differentiation.

To reveal its role in adipogenesis, HMG20A was silenced in SVF cells. The three designed siRNA significantly downregulated HMG20A gene expression (Figure 1D), and sus-siHMG20A-1 was used in subsequent experiments. We found cells transfected with HMG20A siRNA had markedly bigger lipid droplets than cells transfected with control siRNA (Figure 1E). Additionally, the mRNA levels of adipocyte markers CEBPα, PPARγ, and FABP4 were significantly increased in HMG20A-silenced cells (Figure 1F). Overexpression of HMG20A had the opposite effect (Figure 1G–I). These dates indicate that HMG10A silencing promotes adipogenic differentiation into porcine myogenic preadipocytes.

### 2.2. HMG20A Slows Adipogenesis in C3H10T1/2 Mesenchymal Cells

To further determine the effect of HMG20A, we transfected siRNA targeting HMG20A into C3H10T1/2 cells (Figure 2A), and mus-siHMG20A-2 was used in subsequent experiments. We found a drastically increased lipid accumulation in HMG20A knockdown cells after 7 days of adipogenic induction, as indicated by ORO staining (Figure 2B). The mRNA levels of CEBPα, PPARγ, and FABP4, and protein levels of FABP4 were upregulated with siRNA of HMG20A (Figure 2C,D).

Furthermore, we investigated the effect of HMG20A overexpression in C3H10T1/2 cells. After 7 days of adipogenic induction, HMG20A significantly inhibited a mature adipocytic phenotype of C3H10T1/2 cells (Figure 2F). Consistently, three adipogenic differentiation marker genes CEBPα, PPARγ, and FABP4 were significantly downregulated in HMG20A overexpressed cells (Figure 2G,H). Taken together, these results indicated that HMG20A is a negative regulator that reduces the rate of cellular differentiating into adipocytes. This is consistent with the results in porcine myogenic preadipocytes.

### 2.3. Transcriptomics Analysis Indicates MEF2C as a Target of HMG20A

To study the mechanism of HMG20A inhibiting adipogenesis, RNA-seq was used to determine gene expression profile in C3H10T1/2 cells with or without HMG20A knockdown. A total of 271 genes with significant differences were identified, of which 171 were upregulated and 100 were downregulated (Figure 3A). Subsequently, cluster analysis showed that there was little difference between biological repetitions of different samples in the same group, indicating that the samples had good repeatability (Figure 3B). GO and KEGG enrichment analysis showed that HMG20A plays a role in a variety of biological processes and pathways (C-D). Five potential target genes of HMG20A were screened from the differentially expressed genes, which were zinc finger protein 458, fibroblast growth factor 23, CYP2J12, steroid 5 alpha-reductase 1, and myocyte enhancer factor 2C (MEF2C). The expression trend of these five potential target genes in HMG20A knockdown and overexpression cells was confirmed by qRT-PCR (Figure 3E,F). In order to further determine the targeting relationship between HMG20A and MEF2C, we tested the interaction between the MEF2C promoter sequence and HMG20A by CHIP assay (Figure 3G).

### 2.4. SiRNA Targeting MEF2C Promotes Adipogenesis in C3H10T1/2 Cells

The above results demonstrate that the HMG20A may affect adipogenesis by target MEF2C. MEF2C has been reported to be a positive regulator of osteogenesis and muscle formation [28,29]. siRNA targeting MEF2C was shown to be effective (Figure 4A,B). Thus, we investigated whether siRNA targeting MEF2C alone could affect adipogenic differentiation. The MEF2C siRNA enhanced lipid droplets and the expression of the adipogenic differentiation marker genes CEBPα, PPARγ, and FABP4 (Figure 4C,D).

### 2.5. HMG20A and LSD1 Protein Interaction Exists in Adipogenesis

It has been reported that HMG20A binds to the LSD1-CoREST complex and performs a range of biological functions [19]. To investigate whether the HMG20A protein interacts with LSD1, CoIP were performed to detect the presence of HMG20A/LSD1 interaction in 293T model cells. pCMV-FLAG-HMG20A and pEnCMV-KDM1A-3×HA plasmids were constructed and transfected into 293T cells. The date showed that HMG20A interacted with LSD1 reciprocally in 293T cells (Figure 5A). Subsequently, we examined whether HMG20A inhibited adipogenesis by interaction with LSD1. The results showed that knockdown with HMG20A significantly promoted the adipogenesis of C3H10T1/2 cells, while adipogenic ability was significantly inhibited after LSD1 knockdown (Figure 5B–D). Thus, HMG20A knockdown promoted adipogenic differentiation through LSD1.

To further determine the mechanism, CHIP-qPCR and qRT-PCR were used to detect whether HMG20A and LSD1 affected MEF2C promoter methylation modification and mRNA expression level. The results showed that knockdown with HMG20A significantly reduced the demethylation of lysine 4 on histone H3 protein subunit (H3K4me2) in the MEF2C promoter region, and H3K4me2 modification was restored after simultaneous knockdown of HMG20A and LSD1 (Figure 5E). After si-HMG20A was co-transfected with si-LSD1, the inhibition effect of si-HMG20A alone on MEF2C mRNA expression was recovered (Figure 5F). According to the above research, we made a schematic diagram of HMG20A action (Figure 5G).

## 3. Discussion

Past studies have shown that HMG20A plays a role in biological processes in a variety of cells. HMG20A promotes the proliferation of reactive astrogliosis and preserves the neuronal network homeostasis under low grade inflammation [30]. The HMG20A gene is highly expressed in a variety of cancers, which increases the drug resistance of cancer cells and promotes cell migration. Its expression level can be used as a biomarker for cancer prediction [31,32,33]. Whereas, the role of HMG20A in adipocyte growth and development was barely reported. Previous studies have shown that HMG20A is differentially expressed at different stages of differentiation of large white pig myogenic adipocytes [16]. In the current study, the siRNA constructed by us significantly reduced the expression of HMG20A (Figure 1D) and promoted the adipogenic differentiation of SVF cells isolated and cultured in vitro (Figure 1E,F). To determine whether the function of HMG20A is universal, the knockdown and overexpression of HMG20A in C3H10T1/2 cells was found to inhibit the expression of key lipid-forming genes and lipid deposition (Figure 2). The SVF cells used in this study were isolated from the longissimus dorsi muscle of 15-day-old piglets with the ability to differentiate into mature adipocytes, and the C3H10T1/2 cell line is a mouse mesenchymal stem cell, which is commonly used in the study of adipogenesis. The negative regulatory role of HMG20A in adipogenesis was verified in the above two cell models, and it is speculated that HMG20A plays the same role in other species.

In past studies, HMG20A has been found to be involved in a variety of biological functions as a transcription factor [19,20,22,25]. To identify the mechanism and bona fide targets of HMG20A in adipogenic differentiation, RNA-seq was used to search for genes coordinately altered between HMG20A knockdown and normal cells (Figure 3A,B). mRNA levels of MEF2C were found to be promoted by HMG20A (Figure 3C,D), and the CHIP assay confirmed that HMG20A could transcriptionally improve MEF2C expression in C3H10T1/2 cells (Figure 3E). MEF2C is recognized to be involved in myogenesis and may play a role in maintaining muscle cell differentiation and is regulated by a variety of factors [34,35,36]. Meanwhile, MEF2C plays a role in cardiac morphogenesis [37], vascular development [38], B cell induction [39], T cell response [40] and other biological processes. Furthermore, MEF2C may be a key gene in insulin-induced adipocyte differentiation [41]. Our date suggested that MEF2C knockdown promoted the adipogenic differentiation of C3H10T1/2 cells, which was consistent with HMG20A (Figure 4).

LSD1 were showed to carry out a key role in regulating adipogenesis. It was observed that LSD1 knockdown inhibited adipogenesis by increasing H3K9 dimethylation and decreasing H3K4 dimethylation in promoter regions of key genes such as C/EBPα, PPARγ, UCP1, and Wnt signaling elements [26,42,43,44]. The CoIP assay confirmed that HMG20A could bind to LSD1 (Figure 5A). To further determine whether HMG20A plays a role through an interaction with LSD1, co-transfection of si-HMG20A and si-LSD1 could alleviate the promotion effect of HMG20A knockdown alone on adipogenesis (Figure 5B,D). CHIP assay was used to determine H3K4me2 of the MEF2C promoter regulated by HMG20A and LSD1 (Figure 5E). The results suggested that LSD1 could restore the inhibitory effect of HMG20A on adipogenesis, possibly because HMG20A regulated MEF2C promoter H3K4me2 by forming a complex with LSD1 to regulated MEF2C mRNA level.

## 4. Materials and Methods

### 4.1. Cell Culture

Porcine myogenic SVF cells were isolated from longissimus dorsi tissue of healthy 15-day-old Duroc–Landrace–Yorkshire (DLY) piglets. Briefly, the muscle tissue was digested with 0.2% type II collagenase (Solarbio, Beijing, China) solution for 1 h at 37 °C and then separated by centrifugation. Cell suspensions were filtered by a combination of 70-μm and 40-μm mesh filter. Dulbecco’s modified Eagle’s medium (DMEM, Sigam, Aldrich, USA) with fetal calf serum (FCS, 10%, Gibco, Waltham, MA, USA) and penicillin/streptomycin (1%, Sigma Aldrich, St Louis, MO, USA) was used as culture condition and the liquid was changed for two days. C3H10T1/2 cells and 293T cells used in this study were cultured under the same conditions.

### 4.2. Plasmid and siRNA Transfection

The plasmid or siRNA was transfected according to Lipofectamine 2000 (Invitrogen, Carlsbad, CA, USA) instructions, at which time the cells confluent was approximately 70%. Synthetic siRNA oligonucleotides specific for regions in the mouse HMG20A, MEF2C, LSD1, and pig HMG20A mRNA were designed and synthesized by GenePharma (Shanghai, China) and the sequence information is shown in Table 1. The CDS region of pig HMG20A was successfully cloned in the early stage of the experiment, and the length of the CDS region was consistent with that of the PCR amplification (Appendix A). pCMV-HMG20A-flag was used for HMG20A expression in C3H10T1/2 cells, and the pCMV-N-flag empty vector was used as a control. 

### 4.3. Adipogenic Differentiation and OIL Red O Staining

Lipogenesis was induced by “hormone cocktail” method [45]. C3H10T1/2 cells and SVF cells were treated with 1 μmol/L dexamethasone (DEX), 0.5 mmol/L isobutyl-methylxanthine (IBMX), 10 μg/mL insulin, and 100 μmol/L indometacin (IND). After 4 days, the cells were transferred to only 10 μg/mL insulin, and the fluid was changed every 2 days until large lipid droplets appeared. The 293T cells do not require adipogenic differentiation. For Oil red O staining, the cells were washed twice with PBS and fixed in 4% paraformaldehyde for 30 min at room temperature, followed by immersion in 60% isopropyl alcohol for 1 min. The isopropanol was discarded and stained with freshly diluted ORO (Solarbio, Beijing, China) for 10 min. After staining, the cells were washed with PBS and photographed using a microscope (Life Technologies, Carlsbad, CA, USA).

### 4.4. Quantitative Real-Time Polymerase Chain Reaction (qRT-PCR)

Total RNA was extracted from tissues or cells using the Trizol reagent (Life Technologies, Carlsbad, CA, USA) and transcribed into complementary DNA (cDNA) using a first-strand cDNA synthesis kit. Quantitation of the mRNA level by qRT-PCR was performed on a real-time PCR system (Bio-Rad, Richmond, CA, USA) using SYBR Premix Ex Taq II (TransGen Biotech, Beijing, China). β-actin were served as internal controls in pigs and mouse. Appendix A lists all primer sequences of this study. The formula “2^−ΔΔCt^” was used to calculate the mean of the triplicate cycle thresholds (CT) to obtain the relative gene expression level values.

### 4.5. Western Blot and Co-Inmunoprecipitation (CoIP)

Cells were extracted with Radio Immunoprecipitation Assay (RIPA) Lysis Buffer (Beyotime, Shanghai, China) containing a mixture of protease inhibitors. The protein samples were proportioned with 5× loading buffer and denatured at 100 °C for 10 min. The proteins (15 μL) were separated by electrophoresis in 10% polyacrylamide prefabricated SDS gel (Bio-Rad, Richmond, CA, USA) at constant pressure 80 V for 30 min, followed by 120 V for 70 min. The target protein fragment was transferred to nitrocellulose membranes (NC membranes) under constant pressure of 100 V for 90 min at low temperature. Five percent skim milk was used to block the membranes for 1 h, then the membranes were washed several times with PBST. The membrane was incubated with primary antibody diluent at 4 °C overnight. Primary antibodies were: anti-FABP4 (ABclonal, Wuhan, China), anti-MEF2C (ABclonal, Wuhan, China), and anti-β-actin (Bioss, Beijing, China). The next day, the membranes were washed again and incubated with fluorescent secondary antibody diluent at room temperature for 1 h away from light. The exposure was performed using a gel imaging system, and the data were analyzed using Image Studio software, with β-actin as an internal reference, to analyze the strip gray values. For CoIP experiments, protein samples were incubated with IP grade antibodies followed by the pull-down with protein A/G beads for subsequent western blot analyses.

### 4.6. RNA Sequencing (RNA-Seq)

Total RNA was collected from C3H10T1/2 cells transfected with HMG20A siRNA or si-NC for 2 days, respectively. After qualified by quality test (Appendix A), Illumina Novaseq 6000 was sequenced on machine and a large number of raw reads were obtained. Fasdp software was used to quality control the raw data, and clean reads that could be used for subsequent analysis were obtained after removing joint sequences and low-quality reads (Phred quality score ≤ 10). DESeq2 software is used for gene analysis, the difference between the groups to obtain occurred between the two groups of differentially expressed genes, screening criteria for |log2FC| ≥ 0.585 and *p* value < 0.05. The reference genome for this transcriptome sequencing was GRCm39, and 3 biological replicates were set for each treatment. Sequencing data were uploaded to NCBI (accession: PRJNA857188). 

### 4.7. Chromatin Immunoprecipitation (ChIP) Assay

ChIP was performed using the ChIP Assay Kit according to the manufacturer’s instructions (Beyotime, Shanghai, China). The 293T cells were transfected with si-NC, si-HMG20A, and si-HMG20A+si-LSD1 or pCMV-N-flag plasmid and HMG20A overexpression plasmid to prepare chromatin samples. After crosslinking (1% formaldehyde), quenching by addition of 0.125 M glycine, harvesting in SDS lysis buffer, and sonication digestion, the DNA–protein complexes were immunoprecipitated with ChIP grade antibodies against flag (MBL Beijing Biotech, Beijing, China) or AGO2. DNA obtained from the immunoprecipitation was used to detect relative expression by qRT-PCR.

### 4.8. Statistical Analysis

All values are presented as mean ± standard deviation. Data were statistical analyzed using SPSS v11.0 followed by Duncan’s method used for multiple comparisons, and independent sample T test was used for comparison between experimental group and control group. *p* value less than 0.05 indicated significant difference.

## 5. Conclusions

In summary, we found the inhibitory effect of HMG20A on adipogenic differentiation, identified MEF2C as a target gene of HMG20A, and demonstrated that MEF2C plays the same inhibitory effect. Understanding that LSD1 can bind to HMG20A to alleviate its effects and regulate MEF2C expression through epigenetic regulation. Our study develops a regulatory network of adipose differentiation and provides potential therapeutic targets for obesity treatment.

## Figures and Tables

**Figure 1 ijms-23-10559-f001:**
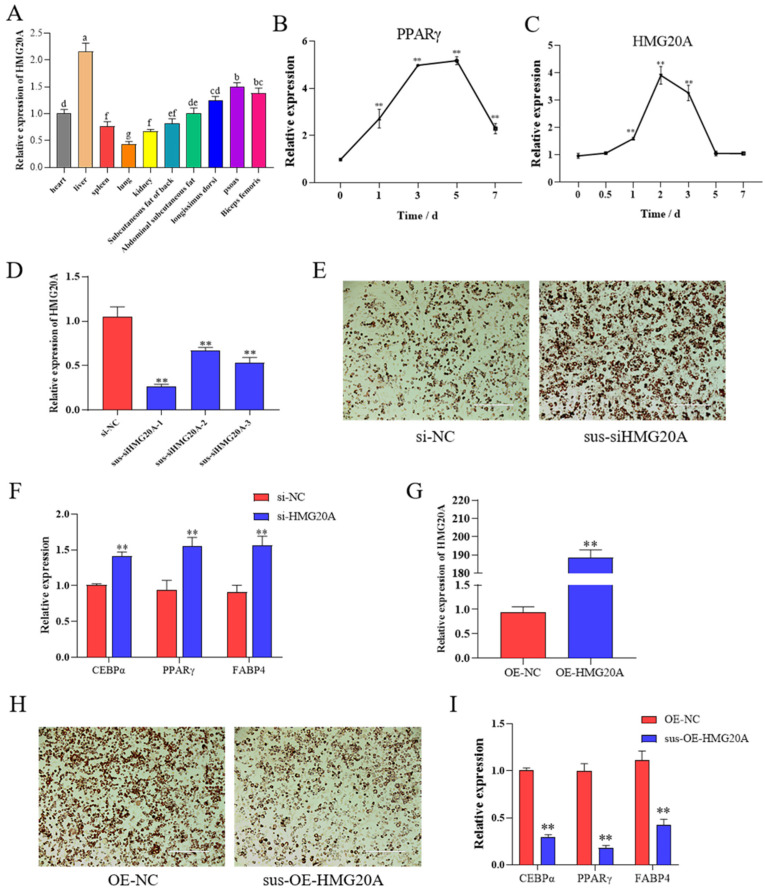
HMG20A knockdown promotes adipogenesis of myogenic SVF cells. (**A**) Expression level of HMG20A in different tissues of Mashen pigs. Bar graphs with the same superscript letters indicate no significant differences (*p* > 0.05), while with different superscript letters indicate significant differences (*p* < 0.05). (**B**,**C**) Timecourse of PPARγ and HMG20A mRNA expression during adipogenic differentiation of SVF cells. SVF cells were differentiated into adipocytes using DMEM, 10% FBS, 0.5mM IBMX, 1mM dexamethasone, 10 mg/mL insulin, and 100umol/L indomethacin. (**D**) qRT–qPCR was used to test the HMG20A silencing efficiency in SVF cells. (**E**) SVF cells were stained with Oil-Red O (ORO) on day 7 after induction of differentiation; magnification: 100×. (**F**) mRNA levels of CEBPα, PPARγ, and FABP4 at day 7 were detected by qRT-PCR. (**G**) qRT–qPCR was used to test the HMG20A overexpression efficiency in SVF cells. (**H**) SVF cells were stained with Oil-Red O (ORO) on day 7 after induction of differentiation; magnification: 100×. (**I**) mRNA levels of CEBPα, PPARγ and FABP4 at day 7 were detected by qRT-PCR. N = 3. Presented as means ± SD (** *p* < 0.01).

**Figure 2 ijms-23-10559-f002:**
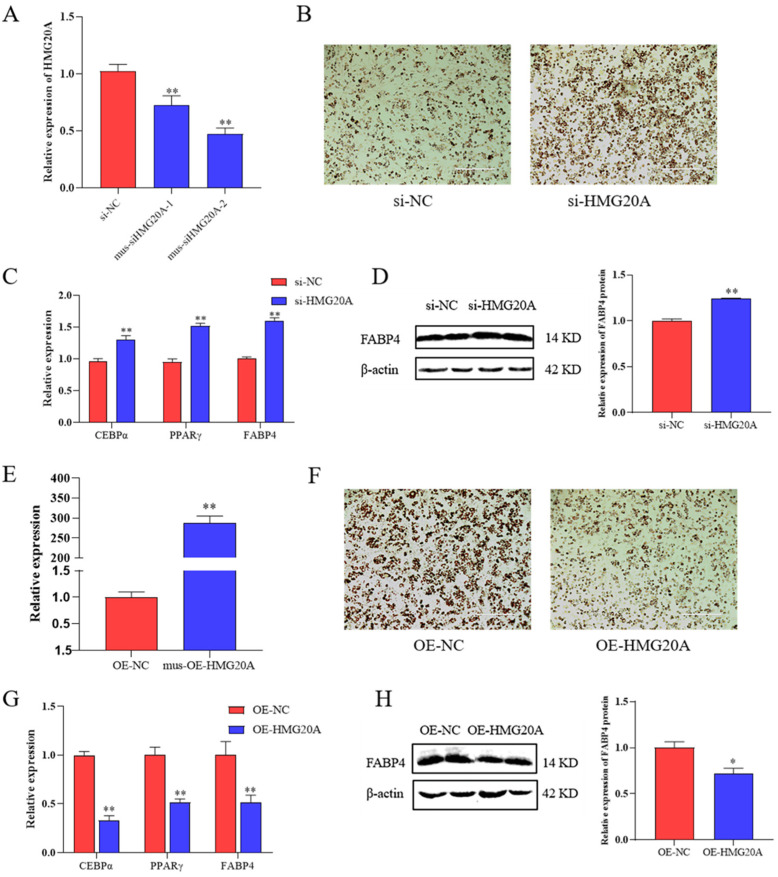
HMG20A inhibited the adipogenesis of C3H10T1/2 cells. (**A**) qRT–qPCR was used to test the HMG20A silencing efficiency in C3H10T1/2 cells. (**B**) C3H10T1/2 cells were stained with Oil-Red O (ORO) on day 7 after induction of differentiation; magnification: 100×. (**C**) mRNA levels of CEBPα, PPARγ, and FABP4 at day 7 were detected by qRT-PCR. (**D**) Protein levels of FABP4 at day 7 were detected by Western blot. The data are represented by grayscale values. (**E**) qRT–qPCR was used to test the HMG20A overexpression efficiency in C3H10T1/2 cells. (**F**) C3H10T1/2 cells were stained with Oil-Red O (ORO) on day 7 after induction of differentiation; magnification: 100×. (**G**) mRNA levels of CEBPα, PPARγ, and FABP4 at day 7 were detected by qRT-PCR. (**H**) Protein levels of FABP4 at day 7 were detected by Western blot. The data are represented by grayscale values. n = 3. Presented as means ± SD (* *p* < 0.05, ** *p* < 0.01).

**Figure 3 ijms-23-10559-f003:**
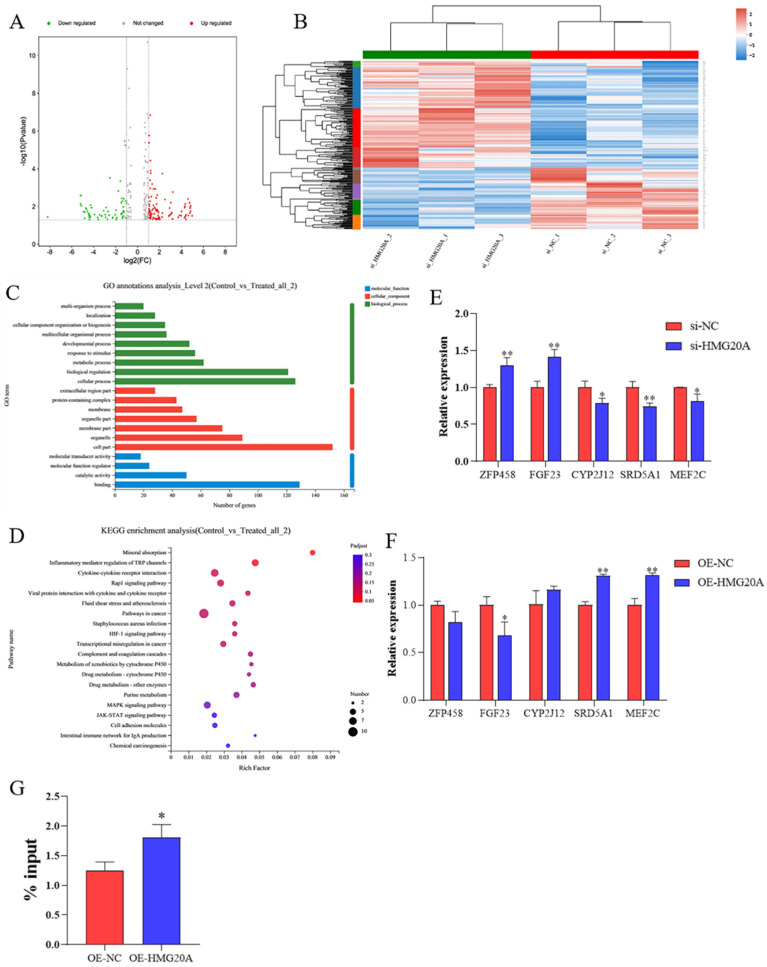
Transcriptomics data. After 7 days of adipogenic differentiation, C3H10T1/2 cells with or without HMG20A knockdown were collected for transcriptome analysis. (**A**) Volcano plot of the differentially expressed genes (DEGs). (**B**) The heatmap of the DEGs. (**C**) GO analysis of DEGs. (**D**) KEGG analysis of DEGs. (**E**,**F**) mRNA levels of 5 candidate target genes in C3H10T1/2 cells that interfere with or overexpress HMG20A. (**G**) ChIP-qPCR analyses of MEF2C reporter levels in C3H10T1/2 cells. Results are presented as ChIP/input DNA ratios (in percent) at the respective promoter regions. n = 3. Presented as means ± SD (* *p* < 0.05, ** *p* < 0.01).

**Figure 4 ijms-23-10559-f004:**
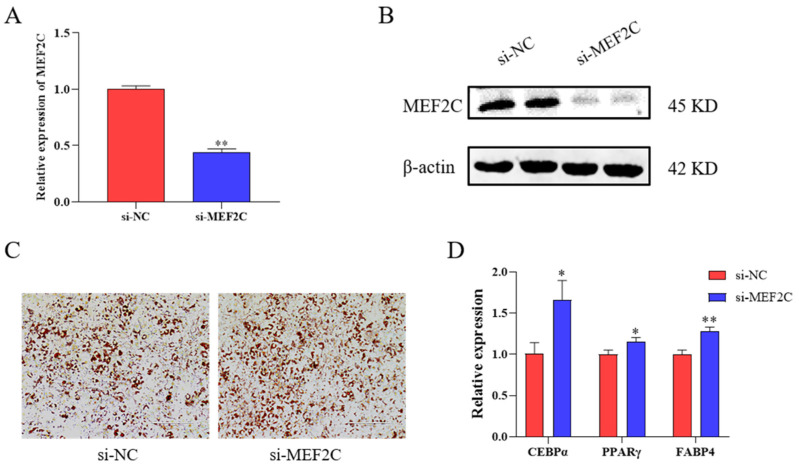
Effects of MEF2C on adipogenic differentiation. (**A**) qRT–qPCR was used to test the MEF2C silencing efficiency in C3H10T1/2 cells. (**B**) Western blot was used to text MEF2C protein levels. (**C**) C3H10T1/2 cells were stained with Oil-Red O (ORO) on day 7 after induction of differentiation; magnification: 100×. (**D**) mRNA levels of CEBPα, PPARγ, and FABP4 at day 7 was detected by qRT-PCR in C3H10T1/2 cells transfected with si-NC and si-MEF2C. n = 3. Presented as means ± SD (* *p* < 0.05, ** *p* < 0.01).

**Figure 5 ijms-23-10559-f005:**
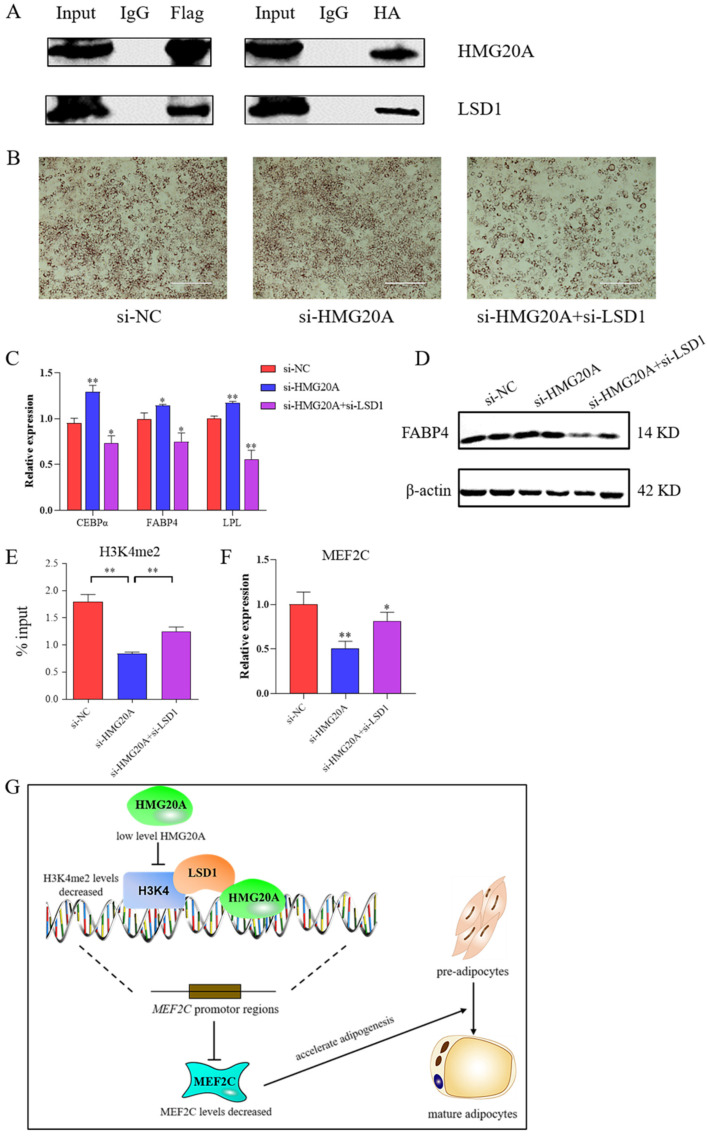
HMG20A and LSD1 protein interaction exists in regulating adipogenesis. (**A**) CoIP of HMG20A and LSD1 co-overexpression C3H10T1/2 cells with anti-Flag or anti-HA antibody followed by HMG20A and LSD1 Western blot. (**B**) C3H10T1/2 cells transfected with si-NC, si-HMG20A and si-HMG20A+si-LSD1 were stained with ORO on day 7 after induction of differentiation. (**C**) mRNA levels of CEBPα, FABP4, and LPL at day 7 was detected by qPCR. (**D**) Protein levels of FABP4 at day 7 was detected by Western blot. (**E**) ChIP-qPCR analyses of H3K4me2 levels of MEF2C reporter in C3H10T1/2 cells. Results are presented as ChIP/input DNA ratios (in percent) at the respective promoter regions. (**F**) mRNA levels of Zfp521 in C3H10T1/2 cells. n = 3. Presented as means ± SD (* *p* < 0.05, ** *p* < 0.01). (**G**) Schematic diagram of HMG20A action.

**Table 1 ijms-23-10559-t001:** The sequence of siRNA.

Sequence Name		Sequence
Negative Control	F:	UUCUCCGAACGUGUCACGUTT
	R:	ACGUGACACGUUCGGAGAATT
sus-siHMG20A-1	F:	CCACUAACAAUCCAGAAUUTT
	R:	AAUUCUGGAUUGUUAGUGGTT
sus-siHMG20A-2	F:	GGAGCGUUACAUGAAGGAATT
	R:	UUCCUUCAUGUAACGGUCCTT
sus-siHMG20A-3	F:	CCCUAUAUUUACAGAGGAATT
	R:	UUCCUCUGUAAAUAUAGGGTT
mus-siHMG20A-1	F:	GACCGUCAGAAAGGCAAAUTT
	R:	AUUUGCCUUUCUGACGGUCTT
mus-siHMG20A-2	F:	GCCUGGAAGUGGAGAAAUATT
	R:	UAUUUCUCCACUUCCAGGCTT
mus-si-MEF2C	F:	CACCUGGUAACCUGAACAATT
	R:	UUGUUCAGGUUACCAGGUGTT
mus-si-LSD1	F:	CCCAAAGAUCCAGCUGACGUUUGAA
	R:	UUCAAACGUCAGCUGGAUCUUUGGG

## Data Availability

All data generated or analyzed during this study are included in this published article. The data that support the findings of this study are available from the corresponding author upon request.

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
