# Peer review of "HMG20A Inhibit Adipogenesis by Transcriptional and Epigenetic Regulation of MEF2C Expression"

_ijms, 2022, doi:10.3390/ijms231810559_

Round 1

Reviewer 1 Report

In this manuscript the authors investigated the role of HMG20A in adipogenesis using SVFs and C3H10T1/2 cells using silencing and looking at downstream effectors and effects. 

Few points.

In Figure 1 the authors show that HMG20A increases with adipogenic differentiation, what is the rationale than to think that it inhibits differentiation?

Are the cells really differentiated with 1 mM Dex or is this a typo? 

Figure 2.  The effects of HMG20A on the expression levels of adipogenic genes in C3H10T1/2 is not very convincing, Neither are the effects on lipid accumulation.

There are some effects on C/EBPalpha (Fig 2G) but the rest is not that great

Figure 4.  When MEF2C is being knocked down, there isn’t really efficient downregulation.  This experiment needs to be repeated. 

It seems that LSD1 may have greater effects. In Figure 5 LSD1 si alone needs to be used as a control.

Also the authors need to address the difference in the models SVF versus C3H10T1/2 in more detail in the discussion.

Reviewer 2 Report

Authors did a time-consuming and intricate work, showing also a great variety of methods, which is a huge advantage of this paper. I am convinced that such a topic will be really interesting for the readers. The biggest advantage is of course an epigenetic theme, which is now of huge interest. 

Therefore, I am quite surprised why authors did not perform studies on human tissue samples. Those results are so catching, that it would be nice to have such, at least initial, data presented herein.

Next, the quality of figures is not a great one. I am pretty sure it could be improved. 

Finally, I am wondering why did the team choose to use only one gene as a control for RT-PCR. Over last years, it is more and more common to use at least two of them, provided both of them are really stable (including screening analysis of at least several control genes during initial steps of preparations for experiments). 

I am fully aware that 18s is a common choice, however, only when none of tested control genes is satisfactory enough. Indeed, there is a huge controversy regarding usage of 18s as a control.  Why did you use it in your model? 

Same question regarding Western Blot analysis - was B-actin abundance compared to other well-known controls before experiments?

Why did you use different controls for mRNA and protein analysis?

Round 2

Reviewer 1 Report

The authors addressed my concerns.

Reviewer 2 Report

Thank you for your response and work made to improve the manuscript. The answers are satisfactory enough for me. However, please continue your research on human cells AND tissues! it would be brilliant to collect human preadipocytes and then perform adipogenesis (it is quite challenging, however woth at least trying). Keep in mind, that especially for human samples the controls should be carefully checked both for Western Blot and RT-PCR. From my experience, I was totally surprised by the outcome of my initial studies regarding stability of expression levels of well-known controls regarding human adipose tissue samples and I have chosen those which were quite rarely used (at least two of them). 

Nice work! Keep going!